# A Comparison of LLM fine-tuning Methods & Evaluation Metrics with Travel Chatbot Use Case

## Abstract

This research compared large language model (LLM) fine-tuning methods, including Quantized Low Rank Adapter (QLoRA), Retrieval Augmented fine-tuning (RAFT), and Reinforcement Learning from Human Feedback (RLHF), and additionally compared LLM evaluation methods including End to End (E2E) benchmark method of "Golden Answers", traditional natural language processing (NLP) metrics, RAG Assessment (Ragas), OpenAI GPT-4 evaluation metrics, and human evaluation, using the travel chatbot use case. The travel dataset was sourced from the Reddit API by requesting posts from travel-related subreddits to get conversation prompts and personalized travel experiences, and augmented for each fine-tuning method. QLoRA and RAFT were applied to two pre-trained LLMs: LLaMa 2 7B and Mistral 7B. The best model according to human evaluation and some GPT-4 metrics was Mistral RAFT, so this underwent a Reinforcement Learning from Human Feedback (RLHF) training pipeline, and ultimately was evaluated as the best model. Our main findings are that: 1) quantitative and Ragas metrics do not align with human evaluation, while Open AI GPT-4 evaluations do, 2) RAFT outperforms QLoRA, but still needs postprocessing, and 3) RLHF improves model performance significantly to outperform benchmark models.

## 1 Introduction

After the COVID-19 pandemic, the influx of travelers was named "revenge travel" to reflect the negative aspects of tourists exceeding the carrying capacity of destinations(Nguwi, 2022). The industry has faced challenges with widespread labor shortages due in short to poor working conditions, which along with the rise of large language model (LLM) applications presents a unique opportunity to incorporate technology into the travel and tourism industry (Binggeli et al., 2023). Travelers prefer

to use technology for the travel from planning to booking to implementation (Peranzo, 2019). With the projected growth of the travel industry post-COVID and recent technological advances, there are potent opportunities for groundbreaking innovation with tangible effects on tourism.

Using the travel use case, this research compared two LLM fine-tuning methods: 1) Quantized Low Rank Adapters (QLoRA) and 2) Retrieval-Augmented Fine-tuning (RAFT). Two pre-trained 7B models, LLaMa 2 and Mistral, are fine-tuned with these two methods, resulting in four models, then their inferences are evaluated against an extensive set of metrics using GPT as a baseline for comparison. The best model is fine-tuned with the third method, 3) Reinforcement Learning from Human Feedback (RLHF), resulting in 5 total models (see Figure 1). The evaluation metrics include: End to End (E2E) benchmark method of "Golden Answers", traditional natural language processing (NLP) metrics, RAG Assessment (Ragas), OpenAI GPT-4 evaluation metrics, and human evaluation.

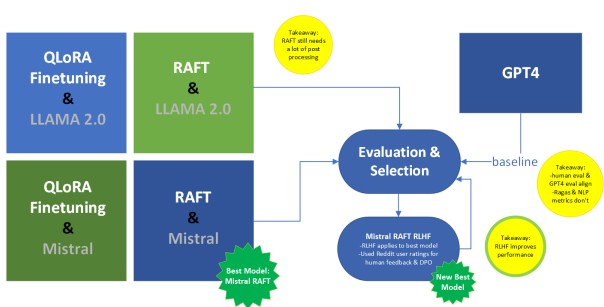

Figure 1: Project Overview with Results and Takeaways

## 2 Related Works

Large language models (LLMs) were made possible with the 2017 "Attention is All You Need" (Vaswani et al.) transformer architecture without recurrent networks or convolutions, and work by predicting masked words or upcoming words (Pahune

and Chandrasekharan, 2023). With LLMs, more data results in better predictions, so most models have at least one billion parameters. Wong et al. (2023) demonstrate how LLMs, particularly Chat-GPT, can revolutionize the tourism industry by enhancing customer experiences in three travel stages: before the trip, en route, and post-trip. ChatGPT improves trip planning efficiency, offers personalized recommendations, and acts as a tour guide or local expert. Despite its advantages, ChatGPT has limitations in accuracy and transparency. It can be biased due to prejudiced training data and limited domain knowledge, leading to misinterpretation of queries. Its data is restricted to the year 2021, so it lacks up-to-date information (Wong et al., 2023). Kang et al. (2023) developed a chatbot using a "tourism information multi-domain DST model and Neo4J graph DB" to provide context-aware personalized travel planner services through transfer learning with a pre-defined multi-domain DST dataset containing tourism data.

Table 1 provides an overview of the selected fine-tuning methods. They were selected because they are open source, have an appropriate size, and have comparable performance. ChatGPT 3.5, considered state of the art, has 154-175 billion parameters. LLaMa 2 7B and Mistral 7B both have 7 billion parameters and, therefore, are faster than Chat-GPT 3.5, require less memory, and performance is slightly inferior, but comparable to ChatGPT 3.5. Mistral 7B outperforms LLaMa 2 on benchmarks like Multi-task Language Understanding (MMLU), TrivialQA, etc. (Mistral AI, 2023; Kang et al., 2023).

In 2023, a highly effective fine-tuning technique called QLoRA was introduced that maintains high accuracy and response quality while reducing the computational and financial cost associated with traditional LLM training. Given the significant computational resources required for fine-tuning LLMs, QLoRA implements several innovative approaches to save memory without compromising performance. It applies gradient back propagation through a frozen 4-bit quantized pre-trained language model, or a subset of the model's most important parameters, into Low Rank Adapters (LoRA), employs double quantization to lower the average memory demand, and incorporates paged optimizers to handle memory spikes (Dettmers et al., 2023). RAG is a text generation method that supplies relevant and factual information from a knowledge base to an LLM (Semnani and Yao, 2023).

Singh (2023) found that if there is a huge corpus of task-specified datasets that have been labeled, fine-tuning is preferable over the retrieval method, RAG, especially for domain-specific tasks, such as specialized topics in the travel domain that lack labeled data.

Wei (2022) introduced chain-of-thought (CoT) prompting as a method to enhance the reasoning ability of large language models (LLMs). This involves inputting a sequence of prompts or instructions to an LLM, which generates text to complete each prompt. CoT prompting improves LLM performance in reasoning by explicitly providing reasoning steps, especially when trained on small datasets (Wei, 2022).

Retrieval Augmented Fine Tuning (RAFT) is a novel method introduced in 2024 to enhance domain-specific Retrieval Augmented Generation (RAG) by adapting pre-trained LLMs (Zhang et al.). RAFT addresses the inability of LLMs to distinguish between context and noise by using data augmentation to create "question, answer, document" triplets. These include oracle documents (verified, relevant documents) and distractor documents. The ideal ratio is 1:4 golden to distractor documents, helping the model learn to extract relevant information using a chain of thought (CoT) reasoning. During inference, RAFT retrieves top-k documents and uses CoT to generate responses, improving performance over standard RAG alone (Zhang et al., 2024). By fine-tuning LLMs with RAFT, the models can generate accurate answers for domain-specific queries by ignoring irrelevant data (Zhang et al., 2024). RAFT can also enable smaller models like LLaMa 2 7B or Mistral 7B to save inference cost and time. The approach is particularly useful for domains with specialized knowledge, like travel, where a curated dataset with questions and answers is used during training.

The Reinforcement Learning from Human Feedback (RLHF) training pipeline for domain-specific LLM curation consists of three main parts: domain-specific fine-tuning, supervised instruction fine-tuning (SFT), and reward modeling (Iyer and Politi, 2023). Reward modeling with RLHF involves training the LLM to classify responses as good or bad using human feedback (Amazon Web Services, 2024). The reward model employed is the Direct Preference Optimization (DPO) trainer, which optimizes preferences using human-annotated triples: prompt, chosen, rejected. The model learns which sentence is more relevant given two options and re-

Table 1: Comparison of Selected Methods

| | QLoRA (Dettmers et al., 2023; Rao, 2023; Singh, 2023) | RAFT (Zhang et al., 2024) | RLHF (Iyer and Politi, 2023; Schmid, 2024) |
|---|---|---|---|
| Characteristics | Fine-tuning of a pre-trained LLM on specific datasets or tasks to achieve desired results | Provide specific instructions within the context of input to elicit a favorable response | Fine-tuning a model with feedback from humans to improve its performance on specific tasks |
| Advantages | Ideal for domain-specific tasks as it is highly adaptable to specific datasets for personalization | Enhance the model's performance in answering questions within specific domains in an "open-book" setting | Improves model alignment with human preferences and enhances the quality of responses through iterative feedback |
| Disadvantages | A huge corpora of data in a specified format is required, specified on tasks related to a certain domain | Very domain specific thus unable to be generalized | Resource-intensive and requires extensive human input for effective training |

quires the dataset to be formatted with the model's template (Schmid, 2024). This process helps align LLM outputs with human preferences, preventing bias and enhancing performance in domain-specific tasks (Iyer and Politi, 2023). Regular re-training helps update models due to observable data drift. The optimal model is one that maximizes the output of the adversarial reward model by accurately classifying responses as good or bad.

Lin and Chen (2023) proposed LLM-EVAL, a unified automatics evaluation method for conversations using LLMs, which is much more simplified and efficient in comparison to current methods using human annotation, ground truth responses, and multiple LLM prompts. LLM-EVAL is a single prompt based evaluation method that uses a unified schema to evaluate conversational quality. Lin claimed that traditional evaluation metrics like and ROGUE are insufficient for natural conversations. LLM-EVAL outperformed other supervised, unsupervised, and LLM-based evaluation metrics

Banerjee et al. (2023) proposed metrics is the End to End (E2E) benchmark method using cosine similarity given a set of predefined answers called the "Golden Answers." The E2E metric compares the chatbot's output results to an expert human answer, or the "Golden Answer." Banerjee et al. and Lin and Chen argue that traditional metrics like Recall-Oriented Understudy for Gisting Evaluation (ROUGE), which uses n-grams overlaps, are insufficient to capture the deep complexity and semantic meaning of a conversational chatbot. Lin

and Chen (2023) makes the same argument including BLEU. E2E is user centric, considers semantic meaning, and improved with advanced prompt engineering alongside human evaluation metrics, unlike ROUGE (Banerjee et al., 2023).

# 3 Travel Datasets Generation

Figure 2 provides a visualization of the data structure needed for each model. The travel dataset was collected entirely from Reddit, and specifically pulled from travel domain subreddits. There are many travel subreddits available, with r/travel being the largest and most active with 8.9 million subscribers. It is one of the most popular communities in the top 1% of subreddits as of December 2023 (red, 2022). Calls made to the Reddit API to curate a corpus of Question-and-Answer (Q&A) formatted data from 201 subreddits. This consists of 27 travel-related subreddits, 30 country subreddits, and 144 city subreddits; example subreddits can be seen in Figure 3. There are a total of 16,300 entries sourced from Reddit, which was further pre-processed and reduced the entries from 16,300 to 10,500 rows by setting a dot score threshold for quality. The data collected for this project was for the purpose of fine-tuning the chatbot, providing domain specific knowledge, and to provide current updated travel information to address LLM knowledge cutoffs.

QLoRA needs structured Question-and-Answer (Q&A) format, but there exists a many to one relationship between a user's Reddit post and com-

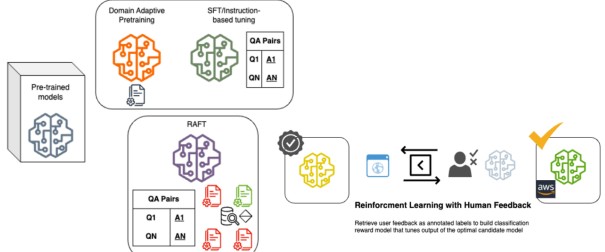

Figure 2: Project Overview and Data Structures

ments from other users in a thread of comments (Rao, 2023). Each Python Reddit API Wrapper (PRAW) request contained several threads with comments ranging from about 90 and over 7400 (see Figure 3). Given the limited context window of LLMs, the entire corpus of information among the subreddits must be partitioned into smaller, and more manageable chunks. To address this issue and ensure a high quality opinionated-based but credible contexts, there are two cutoff parameters: 1) upvote ratio greater than 0.8, which is the number of upvotes over total votes, and 2) first or top 20 comments, which Reddit already has sorted by their internal confidence metrics (PRAW, 2023). Open-source LLM, Falcon 7B, summarized the filtered comments and further removed noise, which resulted in a one to one relationship between question and answer pairs. Finally, the dot score was calculated on the Reddit dataset to find the context relevancy based on the question posted and the Falcon summary, and models were trained on a variety of dot score thresholds.

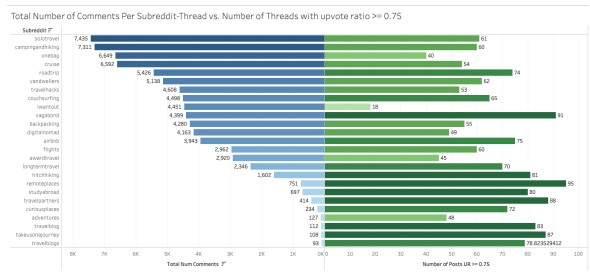

Figure 3: Total Reddit comments per Subreddit and Proportion of "Good" Threads

In RAFT, the training set consists of a question (Q), multiple documents, and an answer that includes a detailed chain of thought (CoT) derived from one of these documents (Zhang et al., 2024). For the travel use case, a RAFT dataset to travel domain specific RAG using Reddit knowledge base was curated, so when given a question and a set of retrieved documents they train the model to only

use relevant documents to answer questions and a specified chunk of irrelevant documents.

RLHF models require human annotated data with good and bad responses (Schmid, 2024). A reinterpretation of of human annotation was applied by selecting the top 10 and bottom 10 percent data, with highest and the worst comments to differentiate among the good and bad response/ accepted or rejected data. Wholistically, the upvote ratio distribution (see Figure 3) across 100 subreddits indicates that posts are generally well-received. It is likely due to collective or personal preference, judgment or bias that results in a low rated thread. The sentiments of each thread, whether good, bad, neutral, or controversial, are reflective of human interaction and behavior and are considered as human annotated data for RLHF.

### 3.0.1 Scientific Artifact Licensing

According to the Data API Terms, the data is owned by the Users, not by Reddit themselves (Dat, 2023). Thus, as long as we do the following, we are allowed to use the Reddit API: are at least 13 years old, are allowed to use the platform according to the judicial system of the United States laws, and follow the seven restrictions of not using Data APIs to harm others, sell information, promote illegal activities, etc. (Dat, 2023). User generated text is likely to contain some personally identifiable information (PII) or profanities (Lee et al., 2023; Lin and Chen, 2023; gua); these have not been filtered out from the data used.

## 4 Model Training & Performance

QLoRA utilizes HuggingFace Transformers, specifically the AutoModelForCausalLM and Auto-Tokenizer pipeline, as well as BitsAndBytesConfig, HfArgumentParser, TrainingArguments, pipeline, and logging (Singh, 2023). QLoRA is quite computationally heavy as it needs a lot of RAM, GPUs, time to train the specific datasets, and memory to store the data. To enable efficient inference generation and save on runtime resources, our trained models are deployed on Huggingface, using AWS as the backed cloud environment for hosting and serving the model. The virtual GPU elected for this deployment is the NVIDIA A10G. Hardware used includes one X64 Lenovo PC, four Apple Macintosh laptops, one high GPU Linux (1x Intel Core i7-6700K 4 GHz Quad-Core, 32GB RAM, 1x 1080 Ti, 1x 1080).

Mistral QLoRA (Figure 4a) had a 0.75 dot score threshold, 286 rows, with an 80/10/10 split. LLaMa QLoRA (Figure 4b) had a 0.65 dot score threshold, 1,425 rows, and a 90/5/5 split. LLaMa RAFT (Figure 4c) had a 0.8 dot score threshold, 420 rows, and a 90/5/5 split. Mistral RAFT (Figure 4d) had 420 rows and a 90/5/5 split. All models had parameters: rank = 64, and alpha = 16. LLaMA RAFT and Mistral RAFT also had parameters double quant = True, batch/eval size = 2, 50 epochs, and 10 and 25 step size respectively.

The datasets were split with an X/Y/Z ratio where X was for training, Y for validation, and Z for testing. Rank is a hyperparameter that improves the performance of the model as it increases the trainable parameters, but it also increases the computational complexity and training time. The four models were trained with the data split and parameters outlined in Table **??**. The table references loss curves for evaluation loss (orange) and training loss (blue). Compared to prior iterations without dot score threshold, the loss curves are improved, with Mistral QLoRA total training loss going from 0.7 to 1.33 (see Figure 4a). LLaMa QLoRA loss at step 25 was 2.86 (see Figure 4b). LLaMa RAFT had a training loss of 1.44 and validation loss of 1.13 at step 25, with overall total loss of 1.29 (see Figure 4c). For Mistral RAFT, the training loss at step 10 was 2.57 and validation loss was 1.97, with a total training loss 2.44 and total validation loss 1.78 (see Figure 4d). Mistral RAFT was selected as the best model for further RLHF training, and the training and evaluation losses converged to zero at step 50 (see Figure 4e).

Due to the format of Reddit data, the prompts are longer than the generic question and answer pair format. Thus, even with 10,000 rows, QLoRA models took over 30 hours to train and fine-tune. However, with hyperparameter tuning this overhead can be reduced by reducing the size of the data by applying dot score thresholds and parameter tuning such as adjusting the step size. Depending on the hyperparameters, training a single epoch ranged from 0.5 to 2.7 hours in the high performance computing lab.

A model deployed on HuggingFace with an inference endpoint runs exceptionally faster than all of the local models with a median latency of 3 seconds and costs a dollar per hour. Locally, pre-trained Mistral took an average of 3,008 seconds to generate an inference, and with the endpoint, 14 seconds. Average inference times ranged from 14

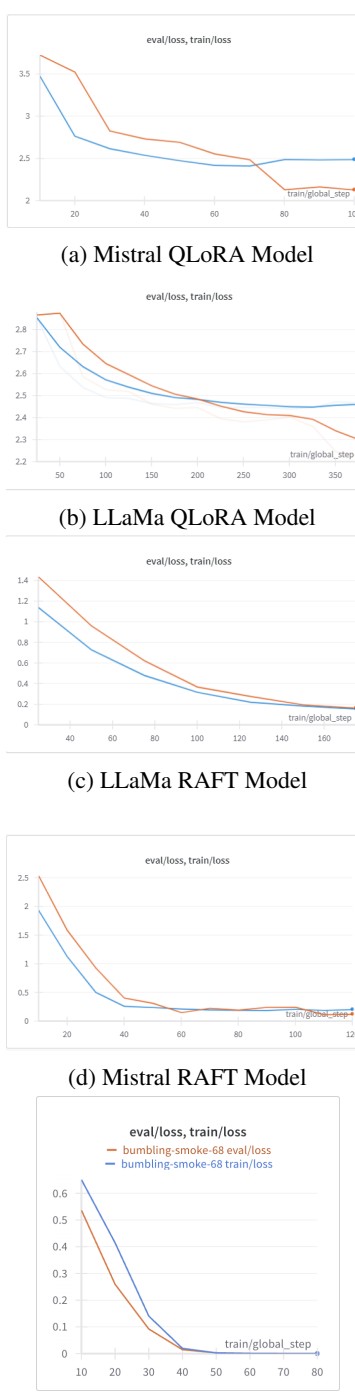

(a) Mistral QLoRA Model

(b) LLaMa QLoRA Model

(c) LLaMa RAFT Model

(d) Mistral RAFT Model

(e) Mistral RAFT RLHF Model

Figure 4: Loss Curves for Different Models (eval-orange, train-blue)

to 44 seconds (see Figure 5). Most models have little variance, including Mistral RAFT which has several outliers. Pre-trained LLaMa and Mistral and Mistral RLHF had a much larger variance in inference time, with pre-trained Mistral clearly as the fastest. This shows the models are not always consistent when generating responses, as it depends on the context, prompt, complexity, and length of

the question.

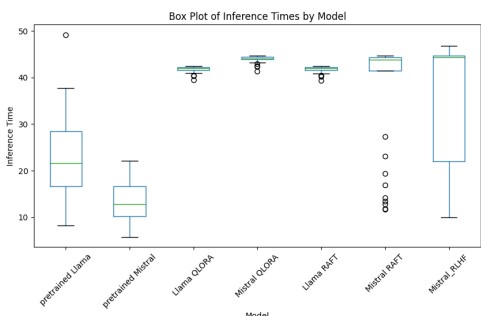

Figure 5: Distribution for simulated runtimes across all fine-tuned and baseline LLMs

# 5 LLM Evaluation Methods

Evaluation of an LLM is different from machine learning (ML) model evaluation due to the complexity of the task involved, unlike ML models, which have more structured prediction tasks. Given this complexity, traditional NLP metrics are not able to adequately evaluate LLMs (Banerjee et al., 2023; Lin and Chen, 2023). LLMs have given a new dimension to artificial intelligence and the evaluation of LLMs is still an ongoing research topic. We have evaluated the aforementioned models against the following quantitative evaluation metrics: 1) Recall-Oriented Understudy for Gisting Evaluation (ROUGE): inexpensive, compares well with human text, measure overlaps of n-grams between generated and reference text, but lacks semantic understanding (Lin, 2004; Chiusano, 2023), 2) ROUGE variants: ROUGE-1, ROUGE-2, ROUGE-L, 3) BertScore: comprises of F1, Precision, and Recall of Bert embeddings of generated and reference text to evaluate semantic similarity (Zhang et al., 2019), 4) BLEU (Bilingual Evaluation Understudy) Score: precision of n-grams in generated and reference text with brevity penalty, 5) Dot Score: Dot product calculates the vector similarity measure that accounts for both magnitude and direction, 6) Cosine Similarity: embedding distance calculates how dissimilar the vectors are; it is the angle of the vector, so the lower the number, the better, and accounts for direction only, and 7) Embedding Distance: LangChain's embedding distance evaluator or cosine distance. All these metrics give the score between 0 to 1, 0 being the worst to 1 being the best response, except for embedding distance, which is the opposite.

Qualitative evaluation metrics include human evaluation, golden answers, and LLM-based evalu-

ation. Human evaluation is a critical and important subjective method of evaluation. Human evaluators gave scores from 0 to 1 with 0.25 increments, 0 being the worst, and 1 being the best. Our evaluation framework relies on the E2E benchmark framework of having golden answers as our ground truth for both the experimental and final evaluation processes. An evaluation dataset of 37 questions is curated from the Reddit dataset, both the golden question and answers are manually curated, and was used to generate inferences from all available models.

LLM-based evaluation included RAG Assessment (Ragas) and Open AI GPT-4 evaluation through LangChain Evaluators. Ragas provides the following metrics to evaluate the two different components–retrieval and generation: contextrelevancy, contextrecall, faithfulness and answerrelevancy. These metrics give the measure of how well the system is retrieving the information, measure of hallucinations and how relevant it is generating the answers with respect to the question (Es, 2023). For this research, only a select subset of LangChain's CriteriaEvalChain was used: coherence, conciseness, helpfulness, and relevance. These return binary values, that are averages across a models' inferences. Like human evaluation, all LLM-based evaluation range from 0 to 1, with higher values indicated better quality of response.

# 6 Results

We have 3 baseline models: pre-trained Mistral, pre-trained LLaMa 2, and GPT-4; 2 Mistral models: Mistral QLoRA, Mistral RAFT; 2 LLaMa 2 models: LLaMa 2 QLoRA, LLaMa 2 RAFT; and RLHF on best model: Mistral RAFT RLHF, for a total of 8 candidate models. There were 27 total metrics: 17 quantitative and 15 qualitative. Each metric has the variance, and was zero for nearly all quantitative metrics. This validates our hypothesis that traditional NLP metrics are insufficient for the complexity of LLMs. Comparing models on metrics with no variance is not useful, so this section is limited to analyzing the nonzero variance metrics (see Figure 6).

Figure 6 indicates the best model, and next best model if the best is a baseline model) and worst performing models highlighted in green and red respectively. Mistral RAFT was selected as the best model, then further fine-tuned with RLHF, so then Mistral RAFT RLHF is the final best model. Mis-

| Model | Average Inference Time (σ²=131.06) | Average Inference PER WORD Time (σ²=0.06) | human eval (σ²=0.04) | faithfulness (σ²=0.09) | answer_relevancy (σ²=0.01) | context_precision (σ²=0.04) | answer_correctness (σ²=0.01) | coherence (σ²=0.10) | conciseness (σ²=0.04) | helpfulness (σ²=0.05) | relevance (σ²=0.01) | GPT4_dot_score (σ²=0.01) | GA_dot_score (σ²=0.12) |
|---|---|---|---|---|---|---|---|---|---|---|---|---|---|
| GPT4 | | | 0.857 | 0.785 | 0.672 | 0.502 | 0.515 | 0.892 | 0.757 | 0.973 | 0.216 | 0.626 | 0.018 |
| pretrained Llama | 22.475 | 0.053 | 0.807 | 0.787 | 0.839 | 0.515 | 0.495 | 1.000 | 0.189 | 0.973 | 0.243 | 0.722 | 0.758 |
| pretrained Mistral | 13.529 | 0.047 | 0.730 | 0.668 | 0.920 | 0.513 | 0.501 | 0.784 | 0.135 | 0.919 | 0.162 | 0.681 | 0.713 |
| Llama QLORA | 41.735 | 0.365 | 0.494 | 0.521 | 0.899 | 0.513 | 0.595 | 0.243 | 0.108 | 0.459 | 0.027 | 0.656 | 0.025 |
| Mistral QLORA | 43.943 | 0.713 | 0.329 | 0.464 | 0.898 | 0.511 | 0.549 | 0.270 | 0.162 | 0.459 | 0.054 | 0.595 | 0.018 |
| Llama RAFT | 41.748 | 0.250 | 0.467 | 0.095 | 0.888 | 0.092 | 0.289 | 0.389 | 0.189 | 0.649 | 0.135 | 0.651 | 0.670 |
| Mistral RAFT | 37.271 | 0.105 | 0.770 | 0.048 | 0.889 | 0.079 | 0.305 | 0.676 | 0.243 | 0.865 | 0.189 | 0.696 | 0.717 |
| Mistral RAFT RLHF | 36.284 | 0.105 | 0.892 | 0.723 | 0.894 | 0.502 | 0.582 | 1.000 | 0.216 | 0.973 | 0.351 | 0.894 | 0.207 |

Figure 6: Evaluation metrics with non zero variance for all models

tral RAFT was the best model by human evaluation of the fine-tuned models, but did not outperform GPT-4 as a baseline, however, Mistral RAFT RLHF did. Mistral RAFT was often next best of the fine-tuned models, including for the following metrics: human evaluation, conciseness, helpfulness, and golden answer dot score, and was selected as the worst for faithfulness and context precision. And yet, human evaluation still determined this model to be the best fine-tuned model, which calls into question the validity of the Ragas and quantitative metrics and highlights the importance of keeping humans in the loop when it comes to evaluation. For human evaluation, Mistral QLoRA the worst; QLoRA models were very repetitive and required post processing. While RAFT models produced the best answers, they also produced multiple answer options and sometimes instructions to a travel agent, which also needed to be removed with post processing.

Looking at Ragas metrics in general, context recall is low across models, answer relevancy very high across models, correctness and precision are of similar values for each respective model, and there is a large variance with faithfulness (see Figure 6). OpenAI GPT-4 metrics most aligned with human evaluation rating Mistral RAFT RLHF and Mistral RAFT as best or next best, and LLaMa QLoRA as the worst, unlike Ragas, which did the opposite. Across the board, coherence and helpfulness are high while conciseness and relevance are generally low. Given the lengthy answers typically generated by LLMs and the manual post processing, which was performed before human evaluation, but not before the OpenAI evaluation, the low values for conciseness make sense. Given the use case of travel recommendation, relevance is arguably the most important metrics, with Mistral RAFT RLHF having the highest value.

Only two quantitative metrics had variance: GPT-4 dot score and golden answer dot score (see Figure 6). GPT-4 dot score found Mistral RAFT RLHF as best and LLaMa QLoRA as the worst.

Golden answer dot score was most aligned with pre-trained LLaMa, Mistral RAFT as next best, LLaMa LLaMa QLoRA as the worst.

Figure 7 shows all the evaluation metrics as a correlation matrix heatmap with red representing a positive correlation and blue representing a negative correlation. Given the disparity of Ragas metrics with human evaluation, we are particularly interested in analyzing the correlations with human evaluation, which has a positive correlation with the following metrics, OpenAI GPT-4 (coherence, consciousness, helpfulness, relevance), and quantitative metrics with no variances: dot scores, ROUGE1, ROUGE2, BLEU, BERT_R, BERT_F1, cosine similarity, and embedding distance. This indicates that OpenAI GPT-4 metrics are most aligned with human evaluation, whereas Ragas metrics are not. While there were many positive correlations with quantitative metrics, there was no variance, indicating these metrics are too simplistic to fully evaluate the complexity of LLMs inference.

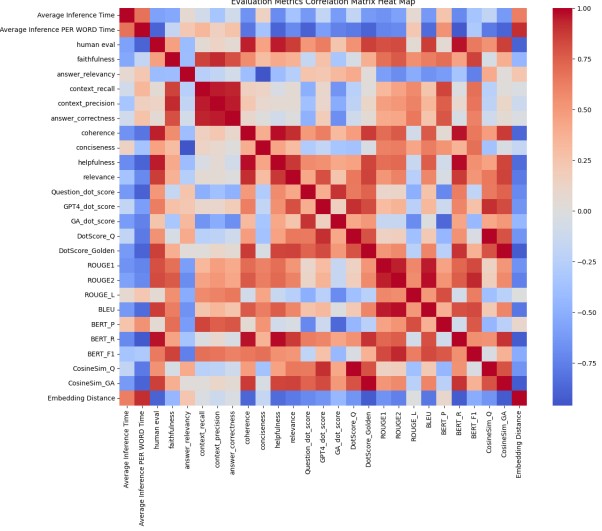

Figure 7: Metrics Correlation Matrix Heatmap

We see that RLHF performance is better than its pre-trained counterpart in all metrics except golden answer dot score and conciseness. RAFT is known to give lengthy answers with multiple an-

swer options, so it is interesting that RLHF further increased this aspect. Conciseness would also impact the dot score as the vector length is assumed to be much longer than the golden answer to which it is being compared.

## 7 Conclusion

With the onset of LLMs, the time required to construct an artificial assistant has been accelerated and the potential for building tailored travel chatbots is becoming more apparent. To meet the demand for a fit-for-purpose chatbot, various implementation strategies for fine-tuning foundational models and evaluation metrics are explored and evaluated.

To conduct this research, a travel dataset was sourced from the Reddit API by querying travel-related subreddits, and then augmented for each fine-tuning method including formats like Q&A format, RAFT, and RLHF. By iteratively transforming for higher quality inputs using dot score threshold, the QLoRA models' ability to learn and generalize improved. Five fine-tuned models: 1) LLaMa 2 QLoRA, 2) Mistral QLoRA, 3) LLaMa 2 RAFT, 4) Mistral RAFT, 5) Mistral RAFT RLHF were publicly deployed on HuggingFace (see Figure 1).

This research compared two fine-tuning methods (QLoRA and RAFT) on foundational LLMs (Mistral and LLaMa 2 7B), finding that Mistral RAFT outperformed the rest. Mistral RAFT was further fine-tuned with RLHF and outperformed all models including baseline models (GPT-4, pre-trained LLaMa, pre-trained Mistral). This refining process yielded an optimal model trained to recognize inputs and contexts in order to deliver inferences aligned with our expectations.

This research also compared LLM evaluation metrics such as the E2E benchmark method, NLP metrics, Ragas, OpenAI GPT-4 evaluation metrics, and human evaluation. While Mistral RAFT RLHF performed best on human evaluation, it performed the worst for some Ragas metrics. Mistral QLoRA performed the worst on most of the quantitative metrics, best on most of the Ragas metrics.

Here are the key findings from our research: 1) Quantitative and Ragas metrics do not align with human evaluation. 2) OpenAI GPT-4 evaluation metrics most closely align with human evaluation. 3) It is essential to keep humans in the loop for evaluation, as traditional NLP metrics are insufficient. 4) Mistral generally outperforms LLaMa. 5) RAFT outperforms QLoRA but still requires post-

processing. 6) RLHF improves model performance significantly.

## 8 Limitations & Risks

Fine-tuning is sensitive to bad data, and despite data cleaning efforts, there is still noise. The models stand to improve the most from better data quality and quantity. Currently, the models may produce a lot of noise and are dependent on the post-processing to parse out unrelated and irrelevant segments that are needed to achieve human-readable outputs. Due to budget and time constraints, comprehensive prompt tuning was not explored and it could be hypothesized that bespoke prompt templates for identified tasks can be pre-written to guide model predictions.

RLHF was conducted on an inferred dataset derived from Reddit ratings in lieu of a direct human rating of the generated text. For human evaluation, it was extremely limited to only three individuals of similar background (female, aged 25-34, Asian), and could benefit from more evaluators. The biggest risk to this research is in the application to the travel use case and having users assume currency of data, even with disclaimers to double check information, since travel related information is real time and ever changing.

## 9 Discussion

A constraint of RAFT is the need for a very diverse set of question-and-answer pairs, and the randomization of ground truth documents is required to help improve performance. RAFT outperformed QLoRA, enabling smaller models like Mistral and LLaMa 2 7B parameter models to save on inference cost, and as hardware continues to evolve, the expenses related to hosting and serving model inference endpoints should reduce. Despite these shortcomings, there remains much potential for mining insights from real-world conversations taking place on online web forums or social media platforms.

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
