# OpenReview forum: "A Comparison of LLM fine-tuning Methods & Evaluation Metrics with Travel Chatbot Use Case"
_ICML.cc/2025/Conference — Submitted to ICML 2025_

### Official Review · Reviewer_mJh4 · 2025-03-03

**Overall Recommendation:** 2

**Summary:**

This paper compares various finetuning and evaluation metrics of LLMs, focusing on the travelling dataset. With RLHF, this paper claims Mistral 7B achieves better performance than GPT-4.

**Claims And Evidence:**

Yes.

**Essential References Not Discussed:**

No.

**Experimental Designs Or Analyses:**

The experiment design is valid.

**Methods And Evaluation Criteria:**

The authors used Reddit dataset to perform RLHF. However, the RLHF utilizes upvote ratio from Reddit, which maybe biased. Although the human evaluators are included in the research, these human evaluators are limited (female, aged 25-34, Asian).

**Other Comments Or Suggestions:**

The authors needs to check the typos in the manuscript. For example, the Table number is not checked in: "The four models were trained with the data split and parameters outlined in Table ??".

**Other Strengths And Weaknesses:**

Strengths: 1.  Explored the different finetuning strategies for LLaMa and Mistral. 2. Used the data from Reddit for the travelling data.
Weakness: 1. The RLHF utilizes upvote ratio from Reddit, which maybe biased. 2. Although the human evaluators are included in the research, these human evaluators are limited (female, aged 25-34, Asian).

**Questions For Authors:**

The authors may consider to include more data sources, which should not exclude to Reddit. Also, the authors may consider to use human labelling score for the RLHF.

**Relation To Broader Scientific Literature:**

This paper explores the different fine-tuning strategies for the travelling dataset. As a result, this paper finds that the RLHF can improve the Mistral 7B model to outperform GPT-4 baseline.

**Theoretical Claims:**

Not applied to this research.

---

### Official Review · Reviewer_bjYi · 2025-03-14

**Overall Recommendation:** 1

**Summary:**

This paper compares various LLMs' fine-tuning methods and evaluation metrics in the context of a travel chatbot. The study evaluates three fine-tuning approaches: QLoRA RAFT, and RLHF, applied to two 7B-parameter LLMs, LLaMa 2 and Mistral. The dataset, sourced from Reddit travel-related subreddits, was augmented into structured formats (e.g., Q&A pairs, RAFT triplets) to support domain-specific training.

**Claims And Evidence:**

The authors summarize six claims in conclusion section, with the first three focusing on human evaluation (e.g., "human evaluation is critical," "GPT-4 metrics align with human judgment"). However, as acknowledged in the "Limitations" section, the human evaluation relied on only three individuals of similar background (female, aged 25–34, Asian). Expanding evaluator diversity (e.g., varying ages, cultural backgrounds, expertise) would strengthen the validity of these claims, as the current limited sample risks biasing results, particularly for culturally sensitive or geographically nuanced travel advice. Additionally, while the authors argue that "traditional metrics are insufficient for LLM evaluation," they do not contextualize this claim within travel-specific challenges. For example: Travel queries often involve dynamic information (e.g., real-time policies, and seasonal events). How well do metrics like Ragas’ "context relevancy" capture such requirements? Such discussions would clarify domain-specific limitations of metrics, moving beyond generic conclusions.
Regarding claims 4–6 (e.g., "RAFT outperforms QLoRA," "RLHF significantly improves performance"), the authors validate method efficacy through metric comparisons but lack travel-domain-specific analysis.  Concrete insights like these would help practitioners prioritize methods for travel use cases, rather than relying on broad assertions like "RLHF works."

**Essential References Not Discussed:**

This job is more like a technical report that reproduces some well-known methods in the travel domain. For this purpose, I don't think there are any references that need to be cited again.

**Experimental Designs Or Analyses:**

The experimental designs and analyses in the paper demonstrate a structured approach but suffer from critical validity issues that weaken confidence in the conclusions.
1. Human evaluation relied on only three evaluators with homogeneous demographics.
2. Most quantitative metrics (e.g., ROUGE, BLEU) showed zero variance across models, but the evaluation used only 37 curated questions. A small dataset may fail to capture the complexity of travel-related queries.
3. The author's analysis is simple and does not revolve around the difficulties in the travel domain. What are the main challenges in deploying fine-tuning methods in travel QA? The author did not provide sufficient insights on this matter.

**Methods And Evaluation Criteria:**

They compare three well-known methods to fine-tune the LLMs. The comparison includes diverse evaluation criteria, including traditional NLP metrics, LLMs scores, and human evaluation.

**Other Comments Or Suggestions:**

typos: missing reference in line 334.

**Other Strengths And Weaknesses:**

The conclusions lack robust validation, as the authors' analysis remains superficial and fails to offer actionable insights into the domain-specific challenges of travel chatbots. For instance, claims about metric-human alignment or RLHF efficacy are inadequately substantiated by empirical evidence, and the limited scope of analysis (e.g., relying on a small, homogeneous evaluation cohort) undermines the generalizability of findings. To strengthen credibility, deeper engagement with travel-specific properties would be necessary.

**Questions For Authors:**

1. Will you release the training dataset and the final model weights?
2. The human evaluation involved only three evaluators of similar demographics (female, 25–34 years, Asian). Could the authors clarify whether they tested for potential cultural or demographic bias in the ratings (e.g., by comparing responses tailored to Western vs. Asian travel preferences)? If not, how do they justify the generalizability of conclusions like "GPT-4 aligns with human judgment"?
3. The authors argue traditional metrics fail to capture LLM complexity but do not propose travel-specific alternatives. Have they explored augmenting metrics with domain-aware criteria (e.g., real-time fact-checking, multi-option support)?
4. The evaluation used only 37 curated questions. How were these questions selected, and what steps were taken to ensure coverage of diverse travel scenarios (e.g., planning, crisis handling, cultural nuances)?

**Relation To Broader Scientific Literature:**

NA

**Theoretical Claims:**

NA

---

### Official Review · Reviewer_BPCs · 2025-03-14

**Overall Recommendation:** 1

**Summary:**

Summary:
The paper mainly focus on the comparison of various fine-tuning methods (QLoRA, RAFT, RLHF). Two pre-trained LLMs (LLaMA 7B & Mistral 7B) were fine-tuned and the performance was evaluated against various metrics. Besides, the author collect the travel dataset from travel-related subreddits and find that Mistral RAFT with further fine-tuned with RLHF, outperformed other models,

Advantages:
1. The practical application on travel chatbot is an interesting case study.
2. The research employs a wide range of evaluation metrics for comparison.

Disadvantages:
1. The article seems to contribute less in terms of comparing LLM fine-tuning methods that are specific to the travel domain, and it is unclear how well the fine-tuning methods and evaluation metrics would generalize to other domains or tasks.
2. Regarding the data acquisition costs for different comparisons, including the time required for fine-tuning (not just inference time), additional information is needed to better assist in selecting the appropriate LLM fine-tuning methods.
3. All the images and tables in the article are too rough and difficult to see clearly. Some references for table appear to be incorrect.

**Claims And Evidence:**

see weakness

**Essential References Not Discussed:**

N/A

**Experimental Designs Or Analyses:**

see weakness

**Methods And Evaluation Criteria:**

see weakness

**Other Comments Or Suggestions:**

N/A

**Other Strengths And Weaknesses:**

N/A

**Questions For Authors:**

see weakness

**Relation To Broader Scientific Literature:**

N/A

**Theoretical Claims:**

see weakness

---

### Official Review · Reviewer_1K6r · 2025-03-16

**Overall Recommendation:** 2

**Summary:**

This paper compares LLM fine-tuning methods (QLoRA, RAFT, RLHF) and evaluation methods (E2E benchmarks, NLP metrics, Ragas, GPT-4 metrics, and human evaluation) using a travel chatbot case. Data was sourced from Reddit and augmented for each fine-tuning method. QLoRA and RAFT were applied to LLaMA2-7B and Mistral-7B, with Mistral-RAFT performing best in human and GPT-4 evaluations. RLHF further improved it, making it the top model.

**Claims And Evidence:**

For QLoRA and RAFT comparison, the experiments results show tha RAFT outperforms QLoRA with Mistral on most metrics but with Llama the advantage of RAFT over QLoRA does not appear to be significant.
For metrics alignment, although it is seen that the correlation differences between various metrics and human evaluation, categorizing quantitative metrics, Ragas metrics, and GPT-4 evaluation metrics, and then quantitatively comparing their correlation differences with human evaluation, would be more convincing.

**Essential References Not Discussed:**

No

**Experimental Designs Or Analyses:**

The experiments of models with finetuning methods shows each model is well trained (shows in Figure4). The experiments of Inference Times with different models (shows in Figure5) seems not very related to the topic or claims. The experiments on based-line models and finetuned models to prove RAFT outperforms QLoRA looks not valid enough, since pretrained models and GPT4 never see the Travel data, to show RAFT is better it may need to compare the finetuning methods on more comparable model models(eg. Llama2 7B,Mistral 7B,Gemma 7B, Qwen 7B)

**Methods And Evaluation Criteria:**

The finetuning methods this work applied (not proposed) make sense for the problem and the dataset constructed in this paper appears to be highly effective and holds significant potential value for research on travel chat LLMs.

**Other Comments Or Suggestions:**

From the perspective of motivation, the author believes that travel chat LLMs will help address the challenges in the travel industry. However, why did the author choose to explore the path of fine-tuning smaller LLMs rather than utilizing large-scale parameter LLMs in combination with methods like RAG or AI agents? Can the former achieve performance comparable to the latter while significantly reducing costs?
This work finds that RLHF significantly improves model performance to outperform benchmark models. Could it be that the dataset or knowledge built in this work has never been exposed to these benchmark models?

**Other Strengths And Weaknesses:**

Strength: This work constructs a broadly themed dataset on travel chat and conducts manual data annotation to support RLHF. These efforts will contribute to the development of LLMs focused on travel chat.
Weakness: The experiment part of this article is not well organized, In the "Model Training & Performance" section, a large portion of the content and images showcase the model's loss curves and inference efficiency. However, this doesn't seem to be closely related to the contributions for claims of this paper.

**Questions For Authors:**

See my comments above

**Relation To Broader Scientific Literature:**

The metrics alignment with human preference study is related to the LLM metric study where researches aim to find better metric for model evaluation to get better LLM performance (eg. In [1] it found that traditional evaluation metrics based on the similarity between outputs and reference answers are also ineffective for questions. )
[1]Zheng L, Chiang W L, Sheng Y, et al. Judging llm-as-a-judge with mt-bench and chatbot arena[J]. Advances in Neural Information Processing Systems, 2023, 36: 46595-46623.

**Theoretical Claims:**

No theoretical claims and proofs have been provided in this work.

---

### Decision · Program_Chairs · 2025-05-01

**Decision:**

Reject

**Comment:**

All reviewers appreciate the importance of benchmarking the fine-tuning methods via a real use case, which is exactly what this work is about. However, there are several useful suggestions raised by reviewers on how to further improve the quality and depth of this work, e.g., quantitatively comparing the correlation between different metrics. Hope the reviews are helpful for the authors.